# Neural network based 3D tracking with a graphene transparent focal stack imaging system

Dehui Zhang [1,3], Zhen Xu [1,3], Zhengyu Huang[1,3], Audrey Rose Gutierrez[1], Cameron J. Blocker [1], Che-Hung Liu[1], Miao-Bin Lien[1], Gong Cheng [1], Zhe Liu[1], Il Yong Chun [2✉], Jeffrey A. Fessler [1✉], Zhaohui Zhong [1✉] & Theodore B. Norris [1✉]

Recent years have seen the rapid growth of new approaches to optical imaging, with an emphasis on extracting three-dimensional (3D) information from what is normally a two-dimensional (2D) image capture. Perhaps most importantly, the rise of computational imaging enables both new physical layouts of optical components and new algorithms to be implemented. This paper concerns the convergence of two advances: the development of a transparent focal stack imaging system using graphene photodetector arrays, and the rapid expansion of the capabilities of machine learning including the development of powerful neural networks. This paper demonstrates 3D tracking of point-like objects with multilayer feedforward neural networks and the extension to tracking positions of multi-point objects. Computer simulations further demonstrate how this optical system can track extended objects in 3D, highlighting the promise of combining nanophotonic devices, new optical system designs, and machine learning for new frontiers in 3D imaging.

[1] Department of Electrical Engineering and Computer Science, University of Michigan, Ann Arbor, MI, USA. [2] Department of Electrical Engineering, University of Hawai'i at Manoa, Honolulu, HI, USA. [3]These authors contributed equally: D. Zhang, Z. Xu, Z. Huang. ✉email: iychun@hawaii.edu; fessler@umich.edu; zzhong@umich.edu; tnorris@umich.edu

Emerging technologies such as autonomous vehicles demand imaging technologies that can capture not only a 2D image but also the 3D spatial position and orientation of objects. Multiple solutions have been proposed, including LiDAR systems[1–3] and light-field cameras[4–7], though existing approaches suffer from significant limitations. For example, LiDAR is constrained by size and cost, and most importantly requires active illumination of the scene using a laser, which poses challenges of its own, including safety. Light-field cameras of various configurations have also been proposed and tested. A common approach uses a microlens array in front of the sensor array of a camera[4,5]; light emitted from the same point with different angles is then mapped to different pixels to create angular information. However, the mapping to a lower dimension carries a tradeoff between spatial and angular resolution. Alternatively, one can use optical masks[6] and camera arrays[7] for light field acquisition. However, the former method sacrifices the signal-to-noise ratio and might need a longer exposure time in compensation; the latter device size could become a limiting factor in developing compact cameras. Single-element position-sensitive detectors, such as recently developed graphene-based detectors[8–10] can provide high speed angular tracking in some applications, but do not provide full 3D information.

To have its highest possible sensitivity to light, a photodetector would ideally absorb all the light incident upon it in the active region of the device. It is possible, however, to design a detector with a photoresponse sufficiently large for a given application, that nevertheless does not absorb all the incident light[11–16]. Indeed, we have shown that a photodetector in which the active region consists of two graphene layers can operate with quite high responsivities, while absorbing only about 5% of the incident light[17]. By fabricating the detector on a transparent substrate, it is possible to obtain responsivities of several A/W while transmitting 80–90% of the incident light, allowing multiple sensor planes

to be stacked along the axis of an optical system. We have previously demonstrated a simple 1D ranging application using single pixel of such detectors[18]. We also showed how focal stack imaging is possible in a single exposure if transparent detector arrays can be realized, and developed models showing how light-field imaging and 3D reconstruction could be accomplished.

While the emphasis in ref. [18] was on 4D light field imaging and reconstruction from a focal stack, some optical applications, e.g., ranging and tracking, do not require computationally expensive 4D light field reconstruction[19,20]. The question naturally arises as to whether the focal stack geometry will allow optical sensor data to provide the necessary information for a given application, without reconstructing a 4D light field or estimating a 3D scene structure via depth map. The simple intuition behind the focal stack geometry is that each sensor array will image sharply a specific region of the object space, corresponding to the depth of field for each sensor plane. A stack of sensors thus expands the total system depth of field. The use of sophisticated algorithms, however, may provide useful information even for regions of the object space that are not in precise focus.

The concept of a focal-stack imaging system based on simultaneous imaging at multiple focal planes is shown in Fig. 1a. In the typical imaging process, the camera lens projects an arbitrary object (in this case a ball-and-stick model) onto a set of transparent imaging arrays stacked at different focal planes. With the sensor arrays having a typical transparency on the order of 90%, sufficient light propagates to all planes for sensitive detection of the projected light field. (Of course the final sensor in the stack need not be transparent, and could be a conventional opaque sensor array). Each of the images in the stack records the light distribution at a specific depth, so that depth information is encoded in the image stack. We can then use neural networks to process the 3D focal stack data and estimate the 3D position and configuration of the object.

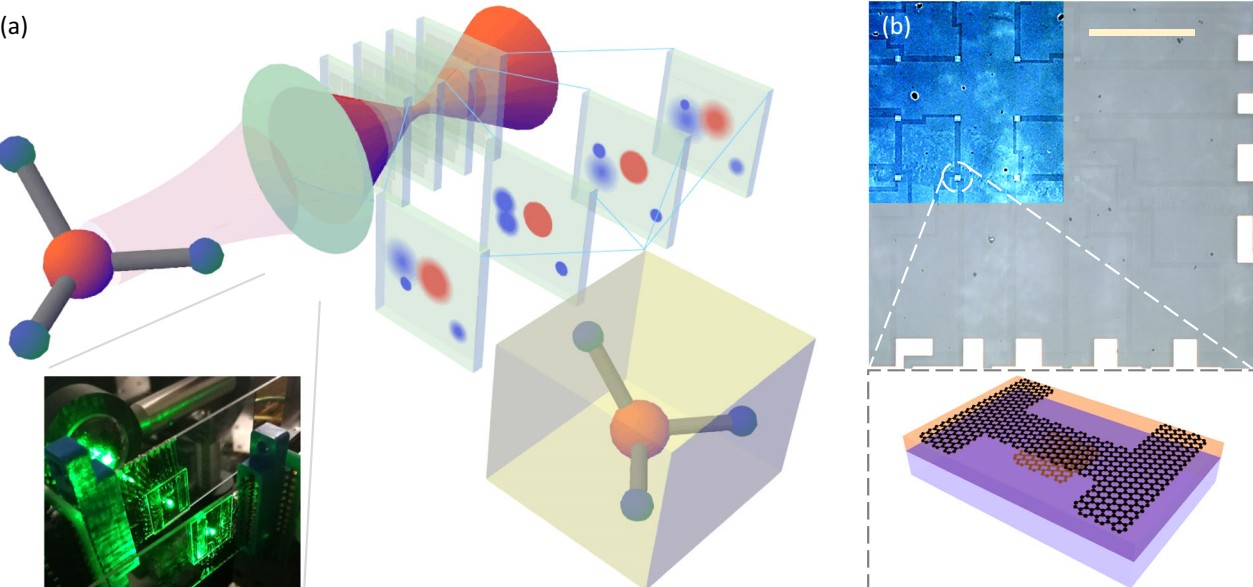

**Fig. 1 Concept of focal stack imaging system enabled by focal stacks of transparent all-graphene photodetector arrays. a** Schematic showing simultaneous capture of multiple images of a 3D object (ball-and-stick model) on different focal planes. Transparent detector arrays (transparent blue sheets) are placed after the lens (green oval) to form the camera system. The depth information is encoded in the image stacks. Artificial neural networks process the image data and extract important 3D configuration information of the object. Inset: photograph of imaging system used in experiments with two transparent focal planes. **b** Upper panel: Optical image of a 4 × 4 transparent graphene photodetector array, scale bar: 500 μm. Upper-left corner is with false color and enhanced contrast in order to highlight the patterns. Lower panel: Schematic of the all-graphene phototransistor design. It includes a top graphene layer as transistor channel and a bottom graphene patch as floating gate, separated by a 6-nm silicon tunneling barrier (purple). The device is fabricated on transparent glass substrate (blue), and the active detector region is wired out with wider graphene stripes as interconnects.

This work demonstrates a transparent focal stack imaging system that is capable of tracking single and multiple point objects in 3D space, without the need for light field reconstruction. The proof-of-concept experiment is demonstrated with a vertical stack of two $4 \times 4$ (16-pixel) graphene sensors and feed-forward neural networks that have the form of a multilayer perceptron (MLP)[21]. We also acquired focal stack data sets using a conventional CMOS camera with separate exposures for each focal plane. The simulations demonstrate the capability of future higher-resolution sensor arrays for tracking extended objects. Our experimental results show that the graphene-based transparent photodetector array is a scalable solution for 3D information acquisition, and that a combination of transparent photodetector arrays and machine learning algorithms can lead to a compact camera design capable of capturing real-time 3D information with high resolution. This type of optical system is potentially useful for emerging technologies such as face recognition, autonomous vehicles and unmanned aero vehicle navigation, and biological video-rate 3D microscopy, without the need for an integrated illumination source. Graphene-based transparent photodetectors can detect light with a broad bandwidth from visible to mid-infrared. This enables 3D infrared imaging for even more applications.

## Results

**All-graphene transparent photodetector arrays**. Photodetector arrays with high responsivity and high transparency are central to realizing a focal stack imaging system. To this end, we fabricated all-graphene transparent photodetector arrays as individual sensor planes. Briefly, CVD-grown graphene on copper foil was wet transferred[22] onto a glass substrate and patterned into floating gates of phototransistors using photolithography. We then sputtered 6 nm of undoped silicon on top as a tunneling barrier, followed by another layer of graphene transferred on top and patterned into the interconnects and device channels. (Fig. 1b bottom inset; Details in Supplementary Information 1) In particular, using an atomically thin graphene sheet for the interconnects reduces light scattering when compared to using ITO or other conductive thin films, which is crucial for recording photocurrent signal across all focal stacks. As a proof-of-concept, we fabricated $4 \times 4$ (16-pixel) transparent graphene photodetector arrays, as shown in Fig. 1b. The active region of each device, the interconnects, and the transparent substrate are clearly differentiated in the optical image due to their differing numbers of graphene layers. The device has an overall raw transparency > 80%; further simulation shows that the transparency can be improved to 96% by refractive index compensation (see Supplementary Information 1). The devices are wired out separately and connected to metal pads, which are then wire-bonded to a customized signal readout circuit. During normal operation, a bias voltage is applied across the graphene channel and the current flowing across the channel is measured; light illumination induces a change in the current, producing photocurrent as the readout (details in Supplementary Fig. 1(a)). The photodetection mechanism of our device is attributed to the photogating effect[17,21,23] in the graphene transistor.

The yield and uniformity of devices were first characterized by measuring the channel conductance. Remarkably, the use of graphene interconnects can still lead to high device yield; 99% of the 192 devices tested show good conductivities (see Supplementary Fig. 1c). The DC photoresponsivity of an individual pixel within the array can reach ~3 A/W at a bias voltage of 0.5 V, which is consistent with the response of single-pixel devices reported previously[18]. We also notice the large device-to-device variation that is intrinsic to most nanoelectronics. Normalization

within the array, however, can compensate for this uniformity issue, which is a common practice even in a commercial CCD array.

To reduce the noise and minimize device hysteresis, the AC photocurrent of each pixel is recorded for 3D tracking and imaging. This measurement scheme sacrifices responsivity but makes the measurement faster and more reliable. As shown in Fig. 2a, a chopper modulates the light and a lock-in amplifier records the AC current at the chopper frequency. The power dependence of the AC photocurrent is also examined (see Supplementary Fig. 1e). The responsivity remains constant in the power range that we use to perform our test. Hence only a single exposure is required to calibrate the nonuniformity between the pixels. We note that the graphene detector speed is currently limited by the charge traps within the sputtered silicon tunneling barrier[17], which can be improved through better deposition techniques and design, as well as higher quality materials[24].

**Focal stack imaging with transparent sensors**. The concept of focal stack imaging was demonstrated using two vertically stacked transparent graphene arrays. As shown in Fig. 2a, two $4 \times 4$ sensor arrays were mounted vertically along the optical axis, separated at a controlled distance, to form a stack of imaging planes. This double-focal-plane stack essentially serves as the camera sensor of the imaging system. A convex lens focuses a 532 nm laser beam, with the beam focus serving as a point object. The focusing lens was mounted on a 3D-motorized stage to vary the position of the point object in 3D. The AC photocurrent is recorded for individual pixels on both front and back detector arrays while the point object is moving along the optical axis.

Figure 2b shows a representative set of images captured experimentally by the two detector arrays when a point object is scanned at different positions along the optical axis (12 mm, 18 mm, 22 mm) respectively, corresponding to focus shifting from the back plane toward the front plane (Fig. 2c). The grayscale images show the normalized photoresponse, with white (black) color representing high (low) intensity. As the focus point shifts from the back plane toward the front plane, the image captured by the front plane shrinks and sharpens, while the image captured by the back plane expands and blurs. Even though the low pixel density limits the image resolution, these results nevertheless verify the validity of simultaneously capturing images at multiple focal planes.

**3D tracking of point objects**. While a single image measures the lateral position of objects as in conventional cameras, differences between images captured in different sensor planes contain the depth information of the point object. Hence focal stack data can be used to reconstruct the 3D position of the point object. Here we consider three different types of point objects: a single-point object, a three-point object, and a two-point object that is rotated and translated in three dimensions.

First, we consider single-point tracking. In this experiment, we scanned the point source (dotted circle in Fig. 2a) in a 3D spatial grid of size $0.6 \times 0.6$ mm ($x$, $y$ axes) $\times 20$ mm ($z$ axis, i.e., the longitudinal direction). The grid spacing was 0.06 mm along the $x$, $y$ axes, and 2 mm along the $z$ axis, leading to 1331 grid points in total. For each measurement, two images were recorded from the graphene sensor planes. We randomly split the data into two subsets, training data with 1131 samples (85% of total samples) and testing data with 200 samples (15% of total samples); all experiments used this data splitting procedure. To estimate three spatial coordinates of the point object from the focal stack data, we trained three separate MLP[25] neural networks (one for each spatial dimension) with mean-square error (MSE) loss. The

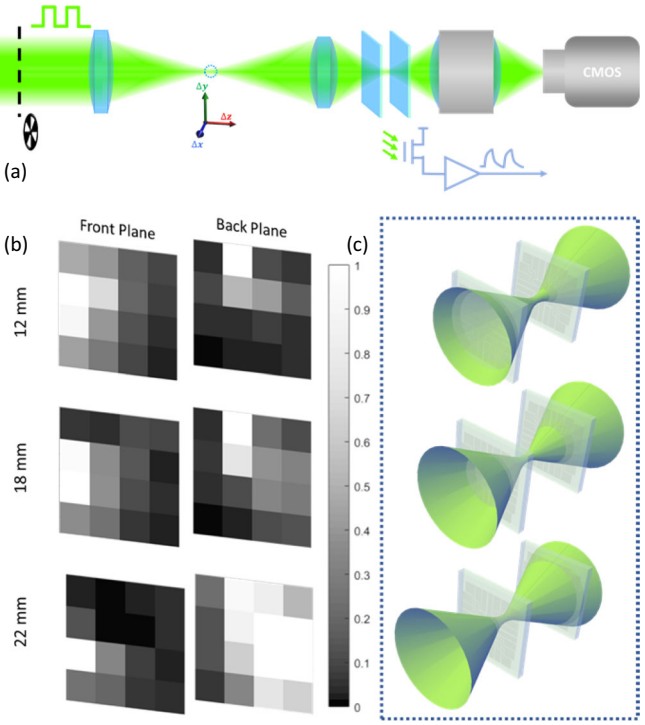

**Fig. 2 Experimental demonstration of focal stack imaging using double stacks of graphene detector arrays. a** A schematic of measurement setup. A point object (dotted circle) is generated by focusing a green laser beam (532 nm) with the lens. Its position is controlled by a 3D motorized stage. Two detector arrays (blue sheets) are placed behind the lens. An objective and CMOS camera are placed behind the detector array for sample alignment. A chopper modulates the light at 500 Hz and a lock-in amplifier records the AC current at the chopper frequency. **b** Images captured by the front and back photodetector planes with objects at three different positions along the optical axis (12 mm, 18 mm, 22 mm, respectively). The grayscale images are generated using responsivities for individual pixels within the array, normalized by the maximum value for better contrast. The point source is slightly off-axis in the image presented, leading to the shift of spot center. **c** The illustrations of the beam profiles corresponding to the imaging planes in (**b**). The focus is shifting from the back plane (top panel) toward the front plane (bottom panel).

results (Fig. 3a, b) show that even with the limited resolution provided by $4 \times 4$ arrays, and only two sensor planes, the point object positions can be determined very accurately. We used the root-mean-square error (RMSE) to quantify the estimation accuracy on the testing dataset; we obtained RMSE values of 0.012 mm, 0.014 mm, and 1.196 mm along the $x$, $y$, and $z$ directions, respectively.

Given the good tracking performance with the small-scale (i.e., $4 \times 4$ arrays) graphene transistor focal stack, we studied how the tracking performance scales with array size. We determined the performance advantages of larger arrays by using conventional CMOS sensors to acquire the focal stack data. For each point source position, we obtained multi-focal plane image stacks by multiple exposures with varying CMOS sensor depth (note that focal stack data collected by CMOS sensors with multiple exposures would be comparable to that obtained by the proposed transparent array with a single exposure, as long as the scene being imaged is static), and down-sampled the resolution of high resolution ($1280 \times 1024$) images captured by CMOS sensor to $4 \times 4$, $9 \times 9$, and $32 \times 32$. We observed that tracking performance improves as the array size increases; results are presented in Supplementary Table 1.

We next considered the possibility of tracking multi-point objects. Here, the object consisted of three point objects, and these three points can have three possible relative positions to each other. We synthesized 1880 3-point objects images as the sum of single-point objects images from either the graphene detectors or the CMOS detectors (see details of focal stack synthesis in Supplementary Information 2). This synthesis approach is reasonable given that the detector response is sufficiently linear and it avoids the complexity of precisely positioning multiple point objects in the optical setup. To estimate the spatial coordinates of the 3-point synthetic objects, we trained an MLP neural network with MSE loss that considers the ordering ambiguity of the network outputs (see Supplementary Information 2, Equation (1)). We used 3-point object's data synthesized from the CMOS-sensor readout in the single-point tracking experiment (with each CMOS image smoothed by spatial averaging and then down-sampled to $9 \times 9$). We found that the trained MLP neural network can estimate a multi-point object's position with remarkable accuracy; see Fig. 3c, d. The RMSE values calculated from the entire test set are 0.017 mm, 0.016 mm, 0.59 mm, along $x$-, $y$-, $z$-directions, respectively. Similar to the single-point object tracking experiment, the multi-point object tracking performance improves with increasing sensor resolution (see Supplementary Tables 2–4).

Finally, we considered tracking of a two-point object that is rotated and translated in three dimensions. This task aims to demonstrate 3D tracking of a continuously moving object, such as a rotating solid rod. Similar to the 3-point object tracking experiment, we synthesized a 2-point object focal stack from single-point object focal stacks captured using the graphene transparent transistor array. The two points are located at the same $x$-$y$ plane and are separated by a fixed distance, as if tied by a solid rod. The rod is allowed to rotate in the $x$-$y$ plane and translate along the $z$-axis, forming helical trajectories, as shown in Fig. 3e. We trained an MLP neural network with 242 training trajectories using MSE loss to estimate the object's spatial coordinates and tested its performance on 38 test rotating trajectories. Figure 3e shows the results of one test trajectory. The neural network estimated the orientation ($x$- and $y$-coordinates) and depth ($z$-coordinate) of test objects with good accuracy: RMSE along $x$-, $y$-, and $z$-directions for the entire test set are 0.016 mm, 0.024 mm, 0.65 mm, respectively.

Supplementary Information 2 gives further details on the MLP neural network architectures and training.

**3D extended object tracking.** The aforementioned objects consisted of a few point sources. For non-point-like (extended) objects, the graphene $4 \times 4$ pixel array fails to accurately estimate the configuration, given the limited information available from such a small array. To illustrate the possibilities of 3D tracking of a complex object and estimating its orientation, we used a ladybug as an extended object and moved it in a 3D spatial grid of size $8.5 \times 8.5 \times 45$ mm. The grid spacing was 0.85 mm along both $x$- and $y$-directions, and 3 mm along $z$-direction. At each grid point, the object took 8 possible orientations in the $x$-$z$ plane, with 45° angular separation between neighboring orientations (see experiment details in Supplementary Information 2). We acquired 15,488 high-resolution focal stack images using the CMOS sensor (at two different planes) and trained two convolutional neural networks (CNNs), one to estimate the ladybug's position and the other for estimating its orientation, with MSE loss and the cross-entropy loss, respectively. Figure 4 shows the results for five test samples. The CNNs correctly classified the orientation of all five samples and estimated their 3D position accurately. For the entire test set, the RMSE along $x$-, $y$-, and

**(a)**

**(b)**

**(c)**

**(d)**

**(e)**

**Fig. 3 3D point object tracking using focal stack data for three different types of point objects. a, b** Tracking results for single point object (only 10 test samples are shown). Results are based on images captured with the graphene photodetector arrays. **c, d** Tracking results for three-points objects (only 4 test samples are shown). Results are based on data synthesized from multi-focal-plane CMOS images (downsampled to 9 × 9) of single point source. **e** Tracking results for rotating two-point objects on one testing trajectory. The object is rotating counter-clockwise (viewed from left) while moving from $z = -10$mm to $z = 10$mm. Results are based on data synthesized from single point source images captured with graphene photodetector arrays.

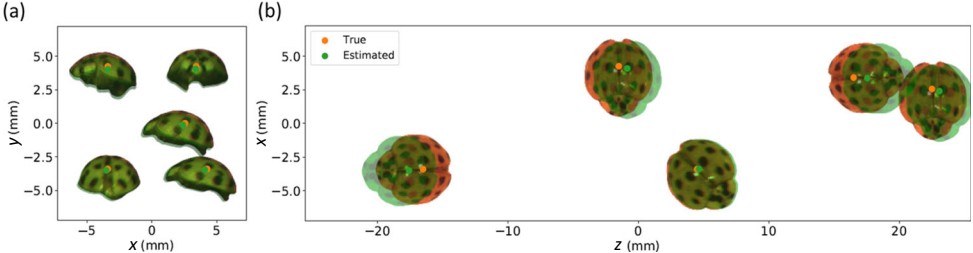

**(a)**

**(b)**

**Fig. 4 3D extended-object tracking and its orientation estimation using focal stack data collected by a CMOS camera. (a)** Results in the *x-y*-plane perspective and **(b)** in the *x-z*-plane perspective. The estimated (true) ladybug's position and orientation are indicated by green (orange) dots and green (orange) overlaid ladybug images. Note that the ladybug images are not a part of the neural network output and are shown for illustration only.

$z$-directions is 0.11 mm, 0.13 mm, and 0.65 mm, respectively, and the orientation is classified with 99.35% accuracy. We note that at least two imaging planes are needed to achieve good estimation accuracy along depth ($z$)-direction: when the sensor at the front position is solely used, the RMSE value along $z$-direction is 2.14 mm, and when the sensor at the back position is solely used, the RMSE value along $z$-direction is 1.60 mm.

Supplementary Fig. 6 describes the CNN architectures and training details.

## Discussion

In conclusion, we designed and demonstrated a focal stack imaging system enabled by graphene transparent photodetector arrays and the use of feedforward neural networks. Even with limited pixel density, we successfully demonstrated simultaneous imaging at multiple focal planes, which can be used for 3D tracking of point objects with high speed and high accuracy. Our computer model further proves that such an imaging system has the potential to track an extended object and estimate its orientation at the same time. Future advancements in graphene detector technology, such as higher density arrays and smaller hysteresis enabled by higher quality tunnel barriers, will be necessary to move beyond the current proof-of-concept demonstration. We also want to emphasize that the proposed focal stacking imaging concept is not limited to graphene detectors alone. Transparent (or semi-transparent) detectors made from other 2D semiconductors and ultra-thin semiconductor films can also be implemented as the transparent sensor planes within the focal stacks. The resulting ultra-compact, high-resolution, and fast 3D object detection technology can be advantageous over existing technologies such as LiDAR and light-field cameras. Our work also showcases that the combination of nanophotonic devices, which is intrinsically high-performance but non-deterministic, with machine learning algorithms can complement and open new frontiers in computational imaging.

## Data availability

The data that support the findings of this study are available from the corresponding authors upon reasonable request.

## Code availability

The code is accessible at https://zenodo.org/record/4282790#.X7gKPshKguU.

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

## Acknowledgements

We gratefully acknowledge financial support from the W. M. Keck Foundation and National Science Foundation grants IIS 1838179. Devices were fabricated in the Lurie Nanofabrication Facility at University of Michigan, a member of the National Nanotechnology Infrastructure Network funded by the National Science Foundation.

## Author contributions

D.Z., Z.X., Z.H., J.A.F., Z.Z., and T.B.N. conceived the experiments. D.Z. and Z.L. fabricated the devices. Z.X., D.Z., C.L., M.L., and G.C. built the optical setup. D.Z., Z.X., and A.R.G. performed the nanodevice optoelectrical measurements. Z.H. performed the CMOS camera data collection. Z.H., I.Y.C., and C.J.B. worked on neural network based 3D reconstructions. All authors discussed the results and co-wrote the manuscript.

## Competing interests

The authors declare no competing interests.
