## [Peer Review File · Nature Communications]

REVIEWER COMMENTS

Reviewer #1 (Remarks to the Author):

1. Authors report Neural Network Based 3D Tracking with a Graphene Transparent Focal Stack Imaging System. Reconstruction algorithms allows simpler layout of optical component systems. Work is continuation of the recently published work of same group in Nature Photonics (Lien M B, Liu C H, Chun I Y, et al. Ranging and light field imaging with transparent 292 photodetectors[J]. Nature Photonics, 2020, 14(3): 143-14) and Nature Nanotechnology (Liu C H, Chang Y C, Norris T B, et al. Graphene photodetectors with ultra-broadband and 312 high responsivity at room temperature[J]. Nature Nanotechnology, 2014, 9(4): 273-278.) Some of the concepts are overlapping with previous above mentioned publications of the same group, so, though authors explored the synergistic design of optical systems in conjunction with image reconstruction algorithms for important application of 3D tracking, I have some reservations on the novelty of this work and I would like the authors to justify it (how it is a significant step forward from the previous two publications). Technical comments on the work are below,
2. Main idea of the work is the 3D tracking of the objects. In principal, tracking speed should be limited by the response time of the photodetector. Since the fundamental mechanism of the photodetectors fabricated in this study is photo gating, so the response time of the photodetectors should be relatively long. Authors did not comment on this aspect. Please mention the speed of the objects in your experiments. Also please measure and report the response time of your detectors.
3. Due to non-ideal transfer and patterning processes which can damage the graphene, reproducibility of the graphene devices is a major issue. An efficient algorithm for the graphene or any nano-devices should be able to detect and compensate for the response of the defective devices in the array.
4. Transparency of the devices is important in this work. One factor that may change the transparency of the devices and can lead to wrong conclusions is that the CVD graphene is not 100% single layer, but has multilayer islands. How do you differentiate the response of the stacks of single layer graphene from the multilayer islands? How will it affect the response normalization process?
5. Please also provide the details of normalization process.
6. What is the percentage coverage of single and multilayer in your CVD graphene? Please provide images showing the graphene layer structure in your devices.
7. How your tracking system is different from CMOS sensors and the other graphene based position detectors?
8. This and the previous two studies of this group (mentioned in the 1st comment) are the proof-of-concept studies. Major challenge (also acknowledged by the authors) in the commercialization the graphene devices is the fabrication the large arrays. So a more logical continuation of the previous two studies should have included the development of the large arrays of focal stacks. Though authors demonstrate a 16 pixel graphene array but it is still not enough for more complex objects.

Reviewer #2 (Remarks to the Author):

In this work, the authors report an interesting graphene based transparent focal stack imaging system. In short, they used multiple transparent graphene photodetectors in the pathway of the imaging formation. Due to the transparency of the photodetectors, multiple images at different position can be captured. Based on these images and a neural network algorithm, the authors are able to extract 3D imagers of the objects.

Overall I think the results are convincing and interesting. In general I feel that this manuscript could be published in Nature Comm. However I do have an important comment on the photodetector performance and would like the authors to clarify it in the revised version. The transparent photodetectors seem to have a very large responsivity in the range of a few A/W. This is extremely high. On the other hand, being transparent indicates that the absorption is weak. This relatively weak absorption but high responsivity indicate that there is significant gain in these

transparent photodetectors. In this case, I think the authors should clarify the gain mechanism, the noise associated with gain, and the sensitivity of the photodetectors. In general, weak absorption is not desirable since any gain will introduce additional noise. As a result, a highly sensitivity photodetector should have strong absorption such that as many "clean" photons as possible can be leveraged.

Reviewer #3 (Remarks to the Author):

"Neural Network Based 3D Tracking with a Graphene Transparent Focal Stack Imaging System" describes a novel image sensor consisting of a 3d array of semi-transparent photo detectors. While many applications can be imagined for such sensors, the authors focus here on 3D object tracking. Of course, since every object point creates a 3D field distribution, true 3D imaging requires correction for the defocused field in each detection plane. The authors accomplish this task using a neural network.

While this is a nice demonstration of the potential utility of 3D focal planes, the sampling model is not very sophisticated and it is not likely that visible imaging systems will actually make use of such functionality. Full focal stack imaging in the visible is relatively easy to implement with swept focus methods, it will be a very long time if ever before 3D sensors will be competitive for this application.

On the other hand, 3D focal planes are of very high potential utility for imaging coherent or partially coherent light, where the spatial sensor distribution could be used for phase sensitive measurement, and for x-ray imaging. It is very likely that this paper will inspire continuing development in this area.

The paper is well written and very clear, I congratulate the authors on their great results and support publication.

David Brady

Response to Referees:

Reviewer #1 (Remarks to the Author):

1. Authors report Neural Network Based 3D Tracking with a Graphene Transparent Focal Stack Imaging System. Reconstruction algorithms allows simpler layout of optical component systems. Work is continuation of the recently published work of same group in Nature Photonics (Lien M B, Liu C H, Chun I Y, et al. Ranging and light field imaging with transparent 292 photodetectors[J]. Nature Photonics, 2020, 14(3): 143-14) and Nature Nanotechnology (Liu C H, Chang Y C, Norris T B, et al. Graphene photodetectors with ultra-broadband and 312 high responsivity at room temperature[J]. Nature Nanotechnology, 2014, 9(4): 273-278.) Some of the concepts are overlapping with previous above mentioned publications of the same group, so, though authors explored the synergistic design of optical systems in conjunction with image reconstruction algorithms for important application of 3D tracking, I have some reservations on the novelty of this work and I would like the authors to justify it (how it is a significant step forward from the previous two publications). Technical comments on the work are below,

We thank the referee for the thoughtful comment, and would like to clarify the novelty and potential impact of the work.

The primary focus of this work is not on the graphene photodetector itself, but rather the attempt of achieving simple yet powerful nanophotonic systems through the coupling of an entirely new optical system design with state of the art machine learning. There is a plethora of papers these days applying neural nets and other machine learning techniques to optical problems. However, it is our belief that genuine breakthroughs using AI methods will especially be enabled by new hardware. In particular, the development of new physical concepts, such as transparent sensor arrays, that might not have been all that promising for real applications due to computational limitations, now become for the first time extremely promising because they can exploit the huge advances in computational power available. The work presented here is one of the first such attempts; indeed, our work demonstrates the ability to acquire optical data in real time in three dimensions (rather than two) by applying neural networks to a completely novel physical system.

On the technical front, the physical device structure used in this work is indeed built upon our previous work (Nature Nanotech 2014, on single devices; Nature Photonics 2020, single devices on separate focal planes), as the referee noted. Although our prior paper did introduce the new conceptual approach to optical imaging system design (focal stacks of transparent detectors), it was clear even at that time that the specific demonstrations presented in that paper were limited by two major challenges. One was that detector *arrays* were not yet available, and the second was that light field reconstruction is computationally intensive, limiting the speed of future applications.

The process of actually making a functional 2D transparent array is challenging and constitutes a significant advance. In the new submission, a 4x4 array is implemented and tested for imaging purposes. Moreover, we proved a high yield of the nanosensors and the graphene interconnects. This paper is the first demonstration of an efficient yet highly transparent photodetector array. This achievement opens up the possibility of higher density arrays, enabling entirely new approaches to the design of optical systems.

The constraint that a compact imaging system has only one focal plane, as we have had from the evolution of the eye to the modern digital camera, is surpassed.

On the algorithm level, the coupling to AI methods overcomes computational limits to its applications. The previous work (Nature Photonics 2020) experimentally implemented 1-D ranging with single pixel detectors on two focal planes and proposed a light field reconstruction algorithm based on regularized linear regression. That algorithm was not implemented for the specific hardware and was intended for a full light field reconstruction. The reconstruction of a single light field takes at least several minutes. And to extract the depth information from the light field (which is often what we want for practical applications), an additional depth estimation step is needed, which takes another several minutes, if a non-learning based method is used.

In this work, we experimentally implemented 3-D point tracking with transparent detector arrays ****without**** requiring a full light field reconstruction. By combining the sensor's 3D sensing capability with the real time tracking algorithm, we bypass the need to do the light field reconstruction and obtain the 3D depth information directly. We leveraged the neural network method to perform object tracking, which, compared to the regularized least squares method, can be time consuming in the training step but can have vast speed up in the inference step, making it more feasible for real-time applications. This is inherently the nature of neural network algorithms.

2. Main idea of the work is the 3D tracking of the objects. In principal, tracking speed should be limited by the response time of the photodetector. Since the fundamental mechanism of the photodetectors fabricated in this study is photo gating, so the response time of the photodetectors should be relatively long. Authors did not comment on this aspect. Please mention the speed of the objects in your experiments. Also please measure and report the response time of your detectors.

The graphene detector arrays used in this work follow the same/similar design as in our previous works (Nature Nanotech 2014), in which we report the detail detector performance including speed. We also discussed in the main text and the supporting information about operating our device at 500 Hz for the 3D ranging and tracking experiment.

We fully agree with the referee that detection speed is important for imaging. Our current generation device has a relatively low speed, which is evident as a drop of responsivity from 3 A/W (DC) to 10 mA/W (Fig. 1S(d)) at 500 Hz chopping frequency. However, the poor gain-bandwidth product is not caused by the photogating mechanism itself. The rich charge traps in the amorphous silicon barrier intercede the interlayer charge transfer and slow down the photoresponse. Photogating effects in such systems can actually produce a decent photoresponse at a much higher frequency: Ref [1] reports improved gain-bandwidth product after HfO₂ passivation. The monolayer MoS₂ phototransistor (also sufficiently transparent) gives a responsivity at 10 A/W at 10 ms response time. A previous work [5] in our group also demonstrates a 60 A/W responsivity at 1 kHz with structure identical to this manuscript, but with Al₂O₃ as the tunneling barrier. Both examples support improved responsivity and speed with a cleaner system. However, we did not employ the Al₂O₃ tunneling barrier in this work, since the few-nanometer-thin Al₂O₃ layer can easily be etched away by the base solution used in the lithography process, which would reduce the device yield. This issue can in principle be addressed with more complex structure and fabrication process designs. Nevertheless, the overall performance of our current device is sufficient as a proof-of-principle demonstration for the multi-focal-plane imaging system.

Experimentally, we chose to operate the device at 500 Hz, given the relatively low but still sufficient responsivity.

We added the following sentence in the revision to highlight the speed limit of the current detector: “**We note that the graphene detector speed is currently limited by the charge traps within the sputtered silicon tunneling barrier (ref 17, 24), which can be improved through better deposition techniques and design, as well as higher quality materials.**”

3. Due to non-ideal transfer and patterning processes which can damage the graphene, reproducibility of the graphene devices is a major issue. An efficient algorithm for the graphene or any nano-devices should be able to detect and compensate for the response of the defective devices in the array.

This is an excellent comment and pinpoints a crucial challenge for implementing nanodevices into applications in general.

First, the reproducibility of the graphene device is quite good given the early stage of development of these devices. In Fig. 1S (b) of the supplementary materials, the channel conductance is measured over 192 devices. We have very few devices showing severely broken channels. The good yield allows us to integrate the detectors into large imaging arrays.

Second, the devices' responsivity variations can be calibrated before the 3D tracking process. Responsivity calibration is a very standard practice even for commercial imagers, as briefly discussed in the manuscript.

Finally, calibrating the responsivities of each pixel is not required for this application. In Fig. 2, we calibrated the image with pixel responsivities to present the right optical power distribution (see captions of Fig. 2(b) for details). However, since the data is then sent to trained neural networks for 3D tracking, the responsivities are already encoded in the weights of neural networks in the training process. This is actually another example that nano-devices benefit from the synergic design with neural networks for realistic applications: *Neural networks can automatically compensate device variations in the training process.*

4. Transparency of the devices is important in this work. One factor that may change the transparency of the devices and can lead to wrong conclusions is that the CVD graphene is not 100% single layer, but has multilayer islands. How do you differentiate the response of the stacks of single layer graphene from the multilayer islands? How will it affect the response normalization process?

The transparency is calibrated with experimental measurements. The reported transparency (81%, no anti-reflection coating) sets a lower limit to the transmittance. The upper limit is around 95% transmission, calculated with proper optical design, as discussed in the supplementary materials.

Moreover, there has been considerable improvement of graphene growth in the past few years. Nowadays, single crystal, monolayer graphene can be fabricated at wafer scale [2]. There is no technical limit that prevents us from building future devices with multilayer-island-free graphene.

We found no trace of multilayer islands with Raman spectroscopy. We transfer a single layer of graphene on a silicon substrate (with 300 nm oxidation layer) with identical process to the fabrication. Gaussian fitting of the 2D band points to a single peak at 2688 cm^{-2} . This, together with the peak ratio over the G-band peak, indicate a uniform phonon resonance condition inside the graphene layer, which matches the

reported spectrum of an exfoliated monolayer [3]. Within the randomly-chosen beam spot region, we did not observe signals from double layer islands in the graphene sample.

5. Please also provide the details of normalization process.

We briefly mentioned the normalization process in the main manuscript. To be more specific, we added the following paragraph to PART A of supplementary materials:

For the responsivity calibration of both detector layers, we first align the lateral position of the imaging chip to maximize the photocurrent readout from the center pixel. This ensures that the optical beam is centered on the chip. Then we move the lens of the camera system along the optical axis to provide a beam spot with a diameter > 4 mm. The large spot size provides nearly uniform illumination in the 0.9-mm-wide detector array. The beam sizes are measured using a power meter and a blade as the moving mask. Then we measure the photocurrent from the array. By calculating the illumination power per device area, we calculate the responsivities of the devices.

6. What is the percentage coverage of single and multilayer in your CVD graphene? Please provide images showing the graphene layer structure in your devices.

The device structure is shown in Fig.1 (b). As stated in the manuscript, each graphene drawn in the graph is monolayer. We have added the following graph to the supplementary materials:

The overlapped channel region (separated by the tunneling barrier) is $10 \mu\text{m}$ by $5 \mu\text{m}$. The lower floating gate layer (green in the figure below) is $20 \mu\text{m}$ by $20 \mu\text{m}$, intentionally made larger to avoid peeling-off. Scale bar: $10 \mu\text{m}$.

7. How your tracking system is different from CMOS sensors and the other graphene based position detectors?

We are not aware of other graphene based position detectors similar to this work. If the reviewer could provide a reference for it, we would be very interested in doing a comparison with it. Having said that, we comment below on the comparison between the proposed system and a system using a single plane CMOS sensor.

The section ‘extended object tracking’ in the main manuscript compares the performance of depth estimation accuracy using multi-focal plane CMOS detector (2 planes) with that using single plane CMOS detector. Experimental results show that a two-plane detector system significantly improves depth estimation accuracy compared to single-plane detector systems. The two-plane detector has an RMSE depth error of 0.65 mm. On the other hand, the single plane detector has an RMSE depth error of 1.6 mm or 2.14 mm, depending on the sensor position.

Our method as presented in the paper is applied at this time only to sparse object tracking and depth estimation. Comparisons with other methods for dense depth maps are beyond the scope of this work.

8. This and the previous two studies of this group (mentioned in the 1st comment) are the proof-of-concept studies. Major challenge (also acknowledged by the authors) in the commercialization the graphene devices is the fabrication the large arrays. So a more logical continuation of the previous two studies should have included the development of the large arrays of focal stacks. Though authors demonstrate a 16 pixel graphene array but it is still not enough for more complex objects.

First, the key innovation of this work is not simply a scaling-up of the previous work, as discussed in the introduction section. Despite being a continuation of the previous multi-focal-plane system, this work implements a different strategy to capture the 3D information using a computationally inexpensive method, rather than reconstructing a full light-field. This provides advantages over the previous approach to many important applications, as discussed in the main manuscript.

Second, the work is already a substantial achievement towards the commercialization of an imaging array. One of the previous works is a study of device physics for a single pixel, which does not at all include scaling-up of the device (Liu C H, et al. Nature Nanotechnology, 2014, 9(4): 273-278.). The other paper, only theoretically proposed the possibility of using such kind of detectors to build arrays, but experimentally did not go beyond single pixels at two of the planes. Even though we only implement two planes with 4x4 pixels on each plane, the 99% yield of the device clearly suggests that we have overcome the challenges set by device and graphene interconnect fabrication. There are of course still crucial

practical challenges toward even larger arrays and ultimately commercialization of the device, however, we view those challenges as technical rather than fundamental and outside of the scope of this work.

Reviewer #2 (Remarks to the Author):

In this work, the authors report an interesting graphene based transparent focal stack imaging system. In short, they used multiple transparent graphene photodetectors in the pathway of the imaging formation. Due to the transparency of the photodetectors, multiple images at different position can be captured. Based on these images and a neural network algorithm, the authors are able to extract 3D imagers of the objects.

Overall I think the results are convincing and interesting. In general I feel that this manuscript could be published in Nature Comm. However I do have an important comment on the photodetector performance and would like the authors to clarify it in the revised version.

The transparent photodetectors seem to have a very large responsivity in the range of a few A/W. This is extremely high. On the other hand, being transparent indicates that the absorption is weak. This relatively weak absorption but high responsivity indicate that there is significant gain in these transparent photodetectors. In this case, I think the authors should clarify the gain mechanism, the noise associated with gain, and the sensitivity of the photodetectors. In general, weak absorption is not desirable since any gain will introduce additional noise. As a result, a highly sensitivity photodetector should have strong absorption such that as many "clean" photons as possible can be leveraged.

We appreciate the referee's comment. In fact, the requirement of having a high responsivity with even a weak absorption is exactly what makes this technique possible and why we choose transparent 2D materials like graphene. The following paragraphs are added to the revised supplementary information:

The noise equivalent power (NEP) is a good measure to discuss the SNR in realistic applications. The NEP of the device has been discussed in the supporting information of our previous work (Liu C H, et al. Nature Nanotechnology, 2014, 9(4): 273-278.). We collect our data with a modulation frequency of 500 Hz. At this frequency, the noise spectral density is 10^{-9} A/Hz^{1/2}. The noise level is consistent with the 1/f noise of graphene transistors observed [4]. This indicates that the channel's 1/f noise dominates over the shot noise of dark currents in the tunneling barrier. With an AC responsivity of 10 mA/W (Fig. 1S (d)), the NEP is 0.1 μ W/Hz^{1/2}. The value is small compared with our test illumination power of ~ 10 μ W per device.

We can also compare this with realistic illumination powers in a camera system. Assume a camera system with a numerical aperture of 0.7. When using it to image a white Lambertian surface under sunlight, the estimated optical power per pixel is 0.05 μ W. This indicates a relatively low SNR for our current device.

The low SNR is largely due to the slow response of our photodetectors, which is caused by the large density of charge traps in the tunneling barrier. Charge traps capture the tunneling charges and compensate the local field that motivates more interlayer hopping. One of our previous work replaced amorphous silicon with high quality Al₂O₃. The responsivity at 1 kHz is as high as 60 A/W at 532 nm [5]. Taking all the corresponding design variations, including increased noise due to a larger channel current, we expect a NEP of 0.1 nW/Hz^{1/2}, which is more than enough for realistic applications. In this experiment, we did not adopt the Al₂O₃ barrier due to fabrication yield considerations, as the thin material

is vulnerable to the base used in lithography. Nevertheless, there are no fundamental limitations that prevent us from fabricating transparent devices with higher speed and responsivity.

In the above discussion, experimental results suggest that the tunneling current is not the major contribution of noise. For a more complete discussion, we can further analyze the tunneling noise's order of magnitude. The shot noise's current spectral density is $S = 2eI$ when the interlayer bias $V \gg kT/e$ [6]. The current is the total of the dark current and the photocurrent, which is around 10 pA in our device. Hence the noise current density of the tunneling photodiode (before amplification) is around $1.8 \text{ fA/Hz}^{1/2}$. The value is much smaller than the photocurrent at any realistic illumination power. Also notice that $1/f$ noise is not considered here, so that the estimation only sets the lower limit for the noise amplitude contributed by the tunneling current before amplified with the photogating effect.

Moreover, neural networks can be trained to be robust against input noises. This further mitigates the SNR requirements for the reported application.

In conclusion, the device's noise is dominated by the $1/f$ noise in the channel. The photoconductive gain amplifies the noise from the vertical tunneling diode. However, it does not dominate the device noise based on both tests and order-of-magnitude estimation. Better implementation of the device to image ambient objects needs an increase in responsivity. One promising way is to improve the tunneling barrier quality, which is also supported by previous work.

Reviewer #3 (Remarks to the Author):

"Neural Network Based 3D Tracking with a Graphene Transparent Focal Stack Imaging System" describes a novel image sensor consisting of a 3d array of semi-transparent photo detectors. While many applications can be imagined for such sensors, the authors focus here on 3D object tracking. Of course, since every object point creates a 3D field distribution, true 3D imaging requires correction for the defocused field in each detection plane. The authors accomplish this task using a neural network.

While this is a nice demonstration of the potential utility of 3D focal planes, the sampling model is not very sophisticated and it is not likely that visible imaging systems will actually make use of such

functionality. Full focal stack imaging in the visible is relatively easy to implement with swept focus methods, it will be a very long time if ever before 3D sensors will be competitive for this application.

We fully agree that there are many aspects to be improved for practical multi-focal-plane imaging systems, especially further scaling up of the imaging arrays and improvement of the device performance. Compared with that, swept focus methods may be temporarily a more mature solution.

However, the swept focus method has its own limitations. Many applications of 3D detection, for example, autonomous driving, demand timely signal processing. The time-division operating scheme of swept focus methods demands a much higher frame rate for capturing multi-focal-plane images. This sets up more challenges on both the readout circuitry as well as the focus-sweeping module, especially for a mechanical one. In contrast, the proposed method allows parallel reading and processing of the image stacks, which leads to a better architecture design. Further improvement of device and readout circuitry will lead to a higher frame rate.

On the other hand, 3D focal planes are of very high potential utility for imaging coherent or partially coherent light, where the spatial sensor distribution could be used for phase sensitive measurement, and for x-ray imaging. It is very likely that this paper will inspire continuing development in this area.

The paper is well written and very clear, I congratulate the authors on their great results and support publication.

Thank you for the feedback.

References

- [1] Kufer D, Konstantatos G. Highly sensitive, encapsulated MoS₂ photodetector with gate controllable gain and speed[J]. Nano letters, 2015, 15(11): 7307-7313.
- [2] Xu X, Zhang Z, Dong J, et al. Ultrafast epitaxial growth of metre-sized single-crystal graphene on industrial Cu foil[J]. Science bulletin, 2017, 62(15): 1074-1080.
- [3] Hao Y, Wang Y, Wang L, et al. Probing layer number and stacking order of few-layer graphene by Raman spectroscopy[J]. small, 2010, 6(2): 195-200.
- [4] Balandin A A. Low-frequency 1/f noise in graphene devices[J]. Nature nanotechnology, 2013, 8(8): 549-555.
- [5] Zhang D, Cheng G, Xu Z, et al. Electrically tunable photoresponse in a graphene heterostructure photodetector[C]//2017 Conference on Lasers and Electro-Optics (CLEO). IEEE, 2017: 1-2.
- [6] Spietz L, Lehnert K W, Siddiqi I, et al. Primary electronic thermometry using the shot noise of a tunnel junction[J]. Science, 2003, 300(5627): 1929-1932.

REVIEWER COMMENTS

Reviewer #1 (Remarks to the Author):

Authors have made most of the asked revisions but haven't provided the requested optical image of the graphene used in devices.

Article may be accepted in its current form.

Authors have asked about some other references related to graphene based position detectors.

Here are some,

1. <https://www.nature.com/articles/lisa2017113/>
2. <https://pubs.acs.org/doi/abs/10.1021/acs.nanolett.9b03368>
3. <https://www.osapublishing.org/optica/fulltext.cfm?uri=optica-5-1-27&id=380597>

If they find it related to their work, I will expect them to compare their position detectors with ones in above mentioned literature sources.

Reviewer #2 (Remarks to the Author):

This is a revised version of the manuscript. The authors addressed my questions and now it can be published

REVIEWER COMMENTS

Reviewer #1 (Remarks to the Author):

Authors have made most of the asked revisions but haven't provided the requested optical image of the graphene used in devices.

Article may be accepted in its current form.

Authors have asked about some other references related to graphene based position detectors.

Here are some,

1. <https://www.nature.com/articles/lisa2017113/>
2. <https://pubs.acs.org/doi/abs/10.1021/acs.nanolett.9b03368>
3. <https://www.osapublishing.org/optica/fulltext.cfm?uri=optica-5-1-27&id=380597>

If they find it related to their work, I will expect them to compare their position detectors with ones in above mentioned literature sources.

The authors gratefully thank the reviewer for the reference papers provided.

The references used a graphene on bulk Schottky heterojunction device to implement photo-detection and used position-dependent response (i.e., the position of the beam spot on the active region) to calibrate for point tracking. Here are some comparisons between our implementation and theirs:

1. By stacking two layers of transparent detector arrays, our implementation can read the depth information. The reference works did not detect out-of-focus blurring or distinguish the direction of motion along the z-axis with their single-pixel detector.
2. In addition to device implementations, a neural network was applied to position tracking in our work. With the rapidly developing algorithms and techniques in deep learning, the pixelized device design coupled with a neural network may be easier to scale and generalize to different tasks or at a different size or speed.
3. While a more complex 3D object tracking is not demonstrated in the reference papers, it is in principle possible with their reported platform. An array of contacts at the edge may be able to extract more features. Combined with a proper reconstruction algorithm, the features might be used for 3D reconstruction of a point object or even an extended complex object. However, such a system down-samples 4D light-field information to 1D edge readouts at the hardware level. This may cause problems in accuracy, resolution, and noise robustness, even with streamlined reconstruction algorithms. On the contrary, we implemented 3D sampling of the light field distribution, demonstrating better accuracy than a 2D sampling (i.e., typical photos). In general, the PSD route does benefit from the compactness, speed, and simple fabrication in certain applications, but also has intrinsic disadvantages that our platform can overcome.

We cited the related papers in our latest manuscript:

Single-element position-sensitive detectors, such as recently developed graphene-based detectors [11-13] can provide high speed angular tracking in some applications, but do not provide full 3D information.

Additional optical microscope image of single pixel is added to the supplementary information:

Fig. 1S (b) Optical microscope image and layout (inset) of a single pixel. Blue: top layer graphene channel; green: bottom layer floating gate. The overlapped channel region (separated by the tunneling barrier) is 30 μm by 10 μm . The lower floating gate layer is 20 μm by 20 μm , intentionally made larger to avoid peeling-off. Scale bar: 20 μm .